# Research on Reduction of Selected Iron-Bearing Waste Materials

**DOI:** 10.3390/ma14081914

**Published:** 2021-04-12

**Authors:** Jan Mróz, Anna Konstanciak, Marek Warzecha, Marcin Więcek, Artur M. Hutny

**Affiliations:** Department of Metallurgy and Metal Technology, Faculty of Production Engineering and Materials Technology, Częstochowa University of Technology, Al. Armii Krajowej 19, 42-200 Częstochowa, Poland; jan.mroz@pcz.pl (J.M.); marek.warzecha@pcz.pl (M.W.); wiecek.marcin@wip.pcz.pl (M.W.); artur.hutny@pcz.pl (A.M.H.)

**Keywords:** waste materials, metal recycling, ironmaking, steelmaking

## Abstract

During the steel production process, nearly twice as many input materials are used as compared to finished products. This creates a large amount of post-production waste, including slag, dust, and sludge. New iron production technologies enable the reuse and recycling of metallurgical waste. This paper presents an investigation on the reduction of selected iron-bearing waste materials in a laboratory rotary furnace. Iron-bearing waste materials in the form of dust, scale, and sludge were obtained from several Polish metallurgical plants as research material. A chemical analysis made it possible to select samples with sufficiently high iron content for testing. The assumed iron content limit in waste materials was 40 wt.% Fe. A sieve analysis of the samples used in the subsequent stages of the research was also performed. The tests carried out with the use of a CO as a reducer, at a temperature of 1000 °C, allowed to obtain high levels of metallization of the samples for scale 91.6%, dust 66.9%, and sludge 97.3%. These results indicate that in the case of sludge and scale, the degree of metallization meets the requirements for charge materials used in both blast furnace (BF) and electric arc furnace (EAF) steelmaking processes, while in the case of reduced dust, this material can be used as enriched charge in the blast furnace process. Reduction studies were also carried out using a gas mixture of CO and H_2_ (50 vol.% CO + 50 vol.% H_2_). The introduction of hydrogen as a reducing agent in reduction processes meets the urgent need of reducing CO_2_ emissions. The obtained results confirm the great importance and influence of the selection of the right amount of reducer on the achievement of a high degree of metallization and that these materials can be a valuable source of metallic charge for blast furnace and steelmaking processes. At an earlier stage of the established research program, experiments of the iron oxides reduction from iron-bearing waste materials in a stationary layer in a Tammann furnace were also conducted.

## 1. Introduction

### 1.1. Quantitative Characteristic s of Metallurgical Wastes

Since 2004, the global crude steel production exceeded the level of one billion tons and indicates the trend of an annual production growth. Crude steel production reached 1869.9 million tons for the year 2019, up by 3.4% compared to 2018 (1808 Mt) [1]. During the steel production process, twice as many input materials are used compared to the finished products. This produces a large amount of post-production waste, namely: blast furnace slag and steel converter slag, blast furnace gas, basic oxygen furnace gas, blast furnace sludge, blast furnace dust, steel sludge, dust from dedusting of a casthouse, sinter sludge and sintering dust, scale and mill scale [2,3,4]. By-products produced by the steel industry are suitable for, either direct or after preparation, re-usage in steelmaking processes and non-metallurgical industry. In the past, the abovementioned post-production waste was usually being stored. New technologies for iron production enable the reuse and recycling of metallurgical waste. Post-production waste materials containing iron oxides and other valuable elements are recycled by being added directly to the metal bath, either by granulating or briquetting, as well as being entered as additional ferrous material in the furnace charge. Metallurgical waste materials can be an alternative to conventional raw materials for construction, cement, and road construction industries [5,6,7]. For economic reasons, metallurgical plants strive to recycle these materials [8]. Interesting research work is presented in papers [9,10,11] on the utilization of waste materials from steel production in electric furnaces. The main assumption of the work was to develop a technology that can be easily implemented “in situ”, i.e., in the conditions of an operating steel plant producing this waste. Moreover, in the work of [9] a method of cementless production of briquettes with the use of fibers from paper recycling as binder was developed that showed a strong effect on the cold compression stability of the agglomerates, by far exceeding other effects such as an increased pressing force.

Table 1 presents the amount of waste materials generated per 1 t production of finished steel in steel mills in Poland. The largest amount are slags: blast furnace slag, steel slag, basic oxygen furnace (BOF) slag. Furthermore, there are scale and dust. The largest amount of sludge comes from the BOF process, and the smallest amount is produced during the blast furnace process.

### 1.2. Qualitative Characteristics of Metallurgical Wastes

Physicochemical properties of slags differ significantly from properties of raw materials used for their production. Slag properties depend on the type of their origin process, the quality of the input materials, and the type of steel being produced [12]. Slags can be classified according to the following groups [13,14]: blast furnace slag, steel slag (including basic oxygen furnace slag—LD slag, electric arc furnace slag—EAF slag, steel secondary refining ladle furnace slag—vacuum arc degassing (VAD), vacuum oxygen decarburization (VOD), ladle furnace (LF), steel continuous casting tundish slag).

Dusts being produced in the course of metallurgical processes arise during the following processes: handling of raw materials for the sintering process, treatment of exhausted gases of iron ore sintering, de-dusting of the crusher, sinter cooling and unloading points, top gas cleaning of blast furnaces, treatment of exhaust gases from oxygen converters, and electric arc furnaces for steel melting. Dusts occurring as the off-gases in these processes have different particle sizes. During mechanical processing of materials, dust with fractions of several tens of micrometers is released; while during thermal changes, evaporation, oxidation, or condensation—particles emitted in the dust—are of the size of several tenths of a micrometer. According to fractional size, dust can be divided into [15]: coarse dust (above 500 μm), average size dust (100 ÷ 500 μm), fine dust (5 ÷ 100 μm), very fine dust (1 ÷ 5 μm), condensed dust, colloidal dust, and smoke (less than 1 μm). Table 2 indicates the percentage of dust being emitted when implementing various technological operations.

Reduction of dust that is emitted to the atmosphere in the exhaust gases is carried out by using dedusting devices, being divided into particular groups [15]: mechanical (dry and wet) dust collectors, fabric or filled filters, and electrostatic (dry or wet) dust collectors. Dust and sludge from sinter plants or blast furnace departments characterized by a high iron content ranging from 20 ÷ 45 wt.% or higher. When assessing the usability and quality of dust, the attention should not only be paid to the iron content but also to other elements, among them carbon. The content of carbon varies and is in the range of 23 ÷ 50 wt.%.

In Poland, approximately 65% of the steel plants based on basic oxygen steelmaking processes use wet dedusting, while electric arc furnace steel plants employ the dry dedusting method. During basic oxygen processes for steel production, dusts and sludge are (calculated on their dry state) produced in the amount of approx. 8 ÷ 12 kg/t of steel production, and in steelmaking processes in electric arc furnaces of—12 ÷ 17 kg/t of steel production [15,16,17].

The stage of the process from which the pollutants originate has a decisive influence on the amount of dust in post-reaction gases. Steelmaking plants in Poland generate the following values of dusts [16]: at the initial stage of the smelting process (0.08 ÷ 3.1 g/m^3^), at the final stage of the smelting process (0.04 ÷ 5.3 g/m^3^), ore refining process (0.07 ÷ 5.5 g/m^3^), oxygen refining process (0.85 ÷ 7.0 g/m^3^), and finishing processes (0.06 ÷ 3.7 g/m^3^). Sludge generated during iron and steel production processes are classified, informal in Poland, according to the following groups: pure iron-bearing sludge (with an iron content exceeding 60 wt.%)—sludge from wet treatment method used for exhaust gases from converter steelworks and wet sludge from scraper troughs of sinter plants, contaminated iron-bearing sludge (with an iron content of 24 ÷ 56 wt.%)—sludge from blast furnace gas treatment, and other sludge, such as sludge from neutralization processes of chemicals or oily mill scale sludge.

Scale and mill scales are formed at high temperatures during oxidation of steel product surfaces as a product of steel reshaping processes. The scale is made by combining iron oxides of various oxidation levels (vustite—FeO, hematite—Fe_2_O_3_ and magnetite—Fe_3_O_4_) with trace amounts of other metal oxides. The chemical composition of scale and mill scale depends on the type of steel and treatment processes being applied. The iron content is approx. 70 wt.%, the rest consists of compounds of various metals and trace amounts of non-ferrous metals and alkali compounds. Oil, grease, and water acting as coolants during treatment processes impede a direct application of the scale [18,19].

Due to the amount of waste being produced in the steel production process, an attempt was made to reduce both indirect and direct iron-bearing waste materials. Representative samples were selected for testing from the collected groups of waste materials, namely dust, scale, and sludge. Research on the reduction of iron oxides for selected ferrous materials with the use of carbon monoxide as a reducing material were carried out. The following industrial waste materials were used [3].

### 1.3. Type of Reducers and Metallurgical Waste

Until recently, the concept of metallurgical waste was mainly associated with solid waste, i.e., slag, dust, and sludge. Currently, it is the gas production of the steel industry and mainly carbon dioxide emissions that are particularly harmful to the environment and climate on a global scale. Of the many proposed methods of reducing carbon dioxide emissions along with carbon capture and utilization technology (CCU) and other technologies, the most effective seems to be the hydrogen as reducing agents. Therefore, in this research work, the aim is to manage selected metallurgical waste. It was decided that one of the reducers would be hydrogen in order to minimize CO_2_ emissions in the process of managing solid metallurgical waste. The second assumption related to hydrogen as a reducing agent is the use of its better reducing properties compared to CO, both in terms of thermodynamic equilibrium and kinetics of the reduction process. Due to the disproportion in particle size (the hydrogen particle diameter is 0.7414 Å, while the carbon monoxide particle diameter is 1.1283 Å), hydrogen indicates greater diffusion capacity. This may affect the kinetics of the reduction process.

At an earlier stage of the established research program, experiments of the iron oxides reduction from iron-bearing waste materials in a stationary layer in a Tammann furnace were also conducted. The findings were presented at the AISTech 2017 annual conference [20].

Currently, not only research works are known, but also industrial implementations of many concepts of using hydrogen in the iron and steel industry are known too [17,18,19,21,22,23,24,25].

## 2. Methodology of Investigations

### 2.1. Sampling

Iron-bearing waste materials in the form of dusts, scale, and sludge were collected as testing material from several metallurgical plants in Poland. A total of five scale samples were obtained: three samples of mill scale, one oily scale, and one scale collected in beds for quenching. A total of 18 dust samples were obtained, including: 6 dust samples from the blasting machine (including lean and coarse fractions), 3 dust samples from casting processes, 3 samples of dust from EAF, 3 samples of grinding dust (from the pipe mill, grinding of castings), one sample of each steel dust from molding and dedusting processes—on the grating shake-out. Four sludge samples were obtained, namely: two samples of sludge from a mixer, a sample of sludge from a blasting machine and a sample of sludge from the field of mill scale. A chemical analysis of the obtained scale, dust, and sludge samples was performed with the LECO CS844 carbon and sulfur analyzer, LECO ONH836 oxygen analyzer, Hitachi S-3400N scanning microscope equipped with Thermo Scientific Noran System 7 EDS detector, and WDS MagnaRay, X’Pert 3 Powder X-ray diffractometer (all these devices are owned by the Laboratories of the Faculty of Materials Science and Engineering of the Silesian University of Technology in Katowice), and chemical “wet” analysis. The chemical analysis allowed selecting for testing samples with sufficiently high iron content. The assumed iron content limit in waste materials for a practical utilization was fixed at >40 wt.% of Fe. A sieve analysis of the samples used in the subsequent stages of the research was also performed [20]. The particle size composition analysis was carried out on woven screens with square mesh sizes: 0.5 mm; 1.0 mm; 2.0 mm; 3.0 mm and 5.0 mm. The dry material was screened on a vibrating mechanical device for 5 min (vibrations 300/min), after which the particular fractions were weighed. A control sieving was then performed for 1 min. 

#### 2.1.1. Chemical and Granulometric Composition of the Scale

The collected samples of the scale met the condition of exceeding 40 wt.% of the total iron content within the range of 72.83 ÷ 85.60 %wt. These were samples 3, 6, 10, 15, 20. The content of the samples included had also insignificant amounts of CaO, SiO_2_, and carbon, as well as other selected components—Table 3 [20].

The summary of the scale sample sieve analysis results is presented in Table 4. It can be noticed that the two smallest fractions of <0.5 mm and 0.5 ÷ 1.0 mm constitute the largest share, as well as the fraction with the largest particles of 5.0 mm (three samples).

#### 2.1.2. Chemical and Granulometric Composition of Dusts

Among the 18 dust samples that were collected, the requirement of 40 wt.% was met by seven samples of waste materials. Table 5 shows the content of carbon, silica, calcium oxide, alkalinity of (CaO/SiO_2_), and other selected components. Particle size composition of dusts is presented in Table 6 [20].

#### 2.1.3. Chemical and Granulometric Composition of Sludge

Four samples of sludge were collected and three of them met the condition of 40 wt.% of iron containing 60, 63 and 78 wt.% of Fe—Table 7. In two samples, a high level of silica content was observed 22.8 and 19.4 wt.% [20].

The particle size composition of the sludge in the dry state is presented in Table 8. For sample # 32, the largest share is within the range of 0.5 ÷ 1.0 mm, which is rare for sludge. Usually, this type of waste receives the highest mass for the smallest fractions in the range of <0.5 mm, as in the cases of samples # 14 and 16.

### 2.2. Methodology

#### Devices Used for Testing

Investigations on reduction processes of iron-bearing waste materials were carried out at research stands at the Department of Metallurgy and Metal Technology at the Częstochowa University of Technology. 

Reduction studies were carried out on a laboratory furnace (FDHK-40/600/1200) with a rotating tube. Properties, such as the type and size of particle fractions, physical properties, and chemical composition of the reduced materials, i.e., mill scale, dusts, and sludge, were assumed when performing the research tests. Comparative testing was carried out for selected samples by using a reduction mixture of 50% CO + 50% H_2_.

The intensification of mixing of the research material in the rotary reaction retort was possible through structural modifications. This was implemented by mounting a ribbed frame of longitudinal slats—blades in the inner part of the retort. During rotation of the reaction tube, blades have been collecting the material by lifting it up to the height of the free fall. This allowed to increase the contact area between the reduced material and the gas. The furnace control system enabled precise maintenance of the set temperature with an accuracy of ±5 °C.

Figure 1 shows the view of the test stand after the upgrade, where in the back one can see the alarm system for controlling the permissible concentration of CO and H_2_ in the environment and the system for gas analysis, recording results, and controlling the furnace (on the left).

### 2.3. Thermochemical Conditions for the Reduction of Iron Oxides by CO and the Mixture (CO + H_2_) in the Range of 200 ÷ 1400 °C

During the indirect reduction of iron oxides, the reducing agent is carbon monoxide, and the reduction reaction proceeds according to the following equation [24]:(1)$$\mathrm{FeO}\;+\;\mathrm{CO}\;=\;\mathrm{Fe}\;+{\;\mathrm{CO}}_{2}\;+\;13.6\;\mathrm{kJ}/\mathrm{mol}$$

The diagram in Figure 2 presenting the Fe-O-C equilibrium system shows that at the temperature of 1000 °C, only about 30% of CO is used for the reduction reaction of FeO to Fe_met_. After that, the system reaches a state of equilibrium.

This means that about 70% by mole volume of CO is chemically inactive towards FeO. The process of reduction to metallic iron reached the level of 30%. In order to achieve full reduction of FeO to metallic iron, the reducing agent in the system should exceed the amount of:(2)$$n=100/{\%\mathrm{CO}}_{2}$$
where n—excessive of CO reducer required for complete reduction of FeO to metallic iron, %CO_2_—equilibrium concentration of CO_2_ in the system at a given temperature, expressed in percent.

For a temperature of 1000 °C, the equation indicates that the excess should be 100/30 = 3.33 of the quantity received from the stoichiometric calculations. Difficulties connected with the exceeding value cease to be significant when the carbon C is present in the system; then the coal gasification reaction (Boudouard reaction) occurs according to the stoichiometric equation [24]:(3)$$C\;+{\;\mathrm{CO}}_{2}\;=\;2\mathrm{CO}\;-\;158\;\mathrm{kJ}/\mathrm{mol}$$

Depending on the carbon reactivity, the complete reaction and the total usage of CO_2_ in the system occurs in the temperature range of 950 ÷ 1000 °C. Under such conditions in the system, CO_2_ is reconstituted into CO, therefore the gas phase composition still has sufficient potential to reduce FeO to metallic iron. This form of reduction by carbon with the formation of CO is called direct reduction [24]:(4)FeO + C = Fe + CO − 144.4 kJ/mol

For the reduction temperature range of 900 ÷ 1050 °C, assumed in this research, the reduction process will take place—both, in the area of mixed and direct reduction. Special advantages of this type of reduction are that there is no need for a large excess of carbon monoxide and that the energy consumption of the entire reduction process is reduced due to the limitation of the scope of direct reduction—the Boudouard reaction is highly energy-consuming.

For a detailed determination of the equilibrium conditions for oxidation-reduction processes in the Fe-C-O and Fe-O-H system for the determined temperature range, computations were performed with the FactSage software package.

Table 9 shows the equilibrium gas composition depending on the temperature. Figure 3 indicates a diagram of the concentration of gaseous reduction products in the temperature range of 850 ÷ 1050 °C.

The course of the equilibrium curves in Figure 3 demonstrates that, as the temperature level increases, the degree of oxidation of the carbon monoxide reduction potential, playing the role of the reducing agent, decreases. Within the designated reduction temperature range of 850 ÷ 1050 °C, the content of equilibrium carbon dioxide as a gaseous product of FeO reduction decreases from 32.55 vol.% to 26.87 vol.%. For hydrogen acting as a reducing agent, the equilibrium curves of the Fe-O reduction demonstrate that an increase in the reduction temperature causes a greater hydrogen consumption in the reduction process. Within the temperature range of 850 ÷ 1050 °C, the equilibrium potential of H_2_O is in the range of 34.17 ÷ 40.19 vol.%. Comparing the properties of the reducers according to the degree of their application in terms of reduction abilities, it can be noticed that at the temperature of 1050 °C the reduction process carried out with the use of hydrogen has an approx. 1.5 times greater reduction potential than the reduction carried out with the use of carbon monoxide.

The research tests were performed with the use of two types of gas reducers: pure carbon monoxide and a gas mixture of 50% CO + 50% H_2_. The amount of reducer was calculated taking into account the chemical composition of waste, the equilibrium conditions for the temperature of 1000 °C, and the reducer excess in the amount of 30% in relation to the amount theoretically necessary for reduction. The flow rate of the reducer was determined from the calculated amount of the reducer and the reduction time of 180 min (this does not apply to additional experiments with a deliberately increased amount of reducer).

When using carbon monoxide, the reduction process was determined continuously, using an automated analysis of CO and CO_2_ in the ABB Uras 14 gas analyzer. This analyzer is part of the Laboratory at the Department of Metallurgy and Metal Technology at the Częstochowa University of Technology. Thus, when using the mixture of CO and H_2,_ the course of the reduction was observed and the amount of water condensing from the post-reaction gases was periodically recorded. After the reduction process, all samples were subject to chemical analysis for the content of total iron (Fe_t_) and metallic iron (Fe_met._). Furthermore, on selected samples, a structural analysis was performed for the content of metallic iron and other iron phases and compounds.

## 3. Results and Discussion

### 3.1. Research Tests on Reduction of Iron Oxides from Ferrous Waste Materials in a Rotary Furnace

In the conducted research testing, the differences in the properties of dusts, mill scale, and sludge were used to investigate the influence of these differences on the course of the reduction process. Thus, the research had a multi-variant form.

#### 3.1.1. Studies on the Reduction of Iron Oxides in Waste Materials by Carbon Monoxide

##### Mill Scale Reduction

Testing on reduction of iron oxides in the mill scale by using carbon monoxide as the reducing agent concerned the problem of assessing the impact of temperature on the reduction process occurring in the rotary furnace. The reduction process was carried out for three temperature variants: I—850 °C, II—900 °C, III—1000 °C. The list of characteristics for particular variants of the testing is presented in Table 10. At the time of heating the furnace within the temperature ranges of 850 ÷ 900 °C and 850 ÷ 1000 °C, a period of non-isothermal reduction occurred, which lasted several minutes. For testing, a sample of the mill scale (Z15) was used, which contained 75.6 wt.% of iron and 0.25 wt.% of coal. The flow rate of carbon monoxide in the reducer was calculated based on the mass of each sample and the volume of oxygen in the sample. Figure 4 shows the course of the reduction based on the amount of condensed water (ml H_2_O/5 min) from post-reaction gases (the values shown are average values from two experiments).

For the temperatures of 900 °C and 1000 °C, the average reduction rates differed slightly and amounted to 0.52 and 0.58 mL of O_2_/min/g (±5%). The temperature of 850 °C is too low to conduct the reduction process effectively—the average reduction rate was 0.15 mL of O_2_/min/g. The obtained results demonstrate that reduction at the temperature of 900 °C is effective without excessive decreasing of the efficiency of the reduction process.

##### Dust Reduction

Reduction of iron oxides in iron-bearing dusts (from foundries) was carried out for two variants: Variant I concerned the usage of various particle sizes (particle size diversity occurs primarily in the finest fraction <0.5 mm), with a similar carbon content—P5A dust contains 98.6 wt.%, while the dust contains 25 ÷ 67.5 wt.% at the same fraction; Variant II (Table 11) regards the use of carbon content diversification in dusts of P5A-3.32 and P11A-0.46 wt.%, with an approximated particle size composition in dusts. For both variants, the reduction process was carried out within the temperature range of 850 ÷ 1000 °C, after reaching the temperature of 1000 °C, reduction was carried out for a period of 120 min. The summary of the reduction conditions for the first variant is shown in Table 12, and the reduction results are shown in Figure 5.

The results of the reduction tests for Variant I showed that the reduction of fine-particle dust (P5A) was faster than that of the medium-particle waste material (P25). The average reduction rate of iron oxides for the P5A dust was 0.33 mL of O_2_/min/g, while for the P25 dust it was 0.20 mL of O_2_/min/g. Finally, the rate of fine dust reduction is 65% higher compared to the medium-particle dust reduction rate.

The research for Variant II concerned the influence of carbon contained in the research material on the course of reduction, because the P5A sample contained 3.32 wt.%. carbon C, while the P11A dust sample contained only 0.46 wt.%. The experimental conditions are summarized in Table 11 and the reduction results are shown in Figure 6.

The presence of carbon in the system significantly increases the rate of reduction, practically in the entire time range, except the initial reduction period—approx. the first 20 min. The average reduction rate for the dust with increased carbon content in the system was 0.26 mL of O_2_/min/g, while for the second dust, with a small amount of carbon, it was 0.19 mL of O_2_/min/g; this proves that the reduction rate is increased by 36.8%. The initial reduction stage, i.e., the first 20 min, with a lower reduction rate for P11A dust, is connected with the mechanism of gradual evolution of the Boudouard reaction—coal gasification.

In the case of the P11A sample, there is a higher reduction rate (and higher carbon content) compared to the P5A sample and the achieved metallization degree is 66.9%.

##### Sludge Reduction

For testing the reduction of iron oxide from sludge, being performed in a rotary furnace, a sludge sample (S16) was used, which was distinguished by a large volume of fine fractions of (<0.5 mm)—81.98 wt.%, high iron content—61.5 wt.%, and low carbon content—1.88 wt.% The main reduction phase with carbon monoxide was performed for 180 min at 1000 °C. The sludge and iron content reduction conditions in the sample before the reduction process are provided in Table 13.

The course of reduction for the sludge samples and the re-testing indicated a high repeatability of the reduction course, and the average values of the reduction rate are very close, namely 68 and 69 mL O_2_/min/g (Figure 7).

Calculation of the oxygen content in iron oxides for samples # 1 (45.52 g) and # 2 (48.25 g) and the oxygen content in the post-reaction gases of 44.1 and 47.49 g, respectively, demonstrate that the reduction degree R, defined as the ratio of the volume of oxygen received and the total amount of oxygen contained in oxides, being expressed in percentage, attains the value of 96.9% for the first sample and 98.4% for the second sample (Figure 8). 

After three hours of reduction, a high degree of reduction was obtained, which indicated an average level of 97.6%. This ensures the degree of metallization required in steelmaking processes.

#### 3.1.2. Studies on the Reduction of Iron Oxides in Waste Materials by the CO + H_2_ Gas Mixture

Investigations on the reduction of iron oxides from iron-bearing waste materials with the use of a gas mixture (50 vol.% of CO + 50 vol.% of H_2_) were carried out for all three groups of materials: dust, mill scale, and sludge. The same waste material types were used in the test samples, namely: scale—15, dust—11A, and sludge—16.

Dust and sludge reduction testing was carried out at 1000 °C in order to use the increased carbon content as a reducing agent.

The reduction of mill scale (15) was carried out in the temperature range of 850 ÷ 1000 °C, with a change every 50 °C in order to determine the impact of temperature on the gas mixture reduction course. The test conditions are summarized in Table 14. The calculation of the volume of reducing gas was with 50% surplus, referring to the needed value to bind the oxygen contained in iron oxides, which is based on the thermochemical equilibrium state at the temperature of reduction.

##### Mill Scale Reduction

The research on the reduction of mill scale was implemented at a temperature range of 850 ÷ 1000 °C. Table 15 demonstrates the conditions for reduction of scale in the samples (15) and Figure 9 shows the course of the reduction process.

Due to little changes in the reduction process at 850 °C, the results were omitted in the description of the temperature conditions of the process.

The samples of the mill scale after reduction were subjected to a chemical analysis for the content of total and metallic irons, which allowed to determine the degrees of metallization for the reducers used (Table 16).

As can be seen from Table 16, depending on the reducer being applied, the difference in degrees of metallization of the reduced scale was 9.4% in favor for the pure carbon monoxide used. The obtained degree of metallization of the material is not suitable for application in the steelmaking process, however, it meets the conditions required for enriched blast furnace charge.

Taking into account the relatively low degree of metallization, it was decided to conduct additional tests with an increased amount of reducing agent. The results of these studies are shown in Table 17.

The data presented in the table show that it is possible to achieve high levels of metallization with the use of a sufficiently large amount of reducer. This qualifies the reduced material for use in steelworks. This means that under these experimental conditions, the amount of reducing agent is the factor that determines the reduction results, i.e., the reduction and metallization.

The obtained results for the mill scale reduction tests with the use of pure carbon monoxide and a mixture (50 vol.% CO + 50 vol.% H_2_) indicate that in the case of a “normal” amount of reducer, i.e., resulting from the chemical composition and equilibrium state of the reaction, better reduction results are obtained when using 100% CO (Table 16) than when using a gas mixture (50 vol.% CO + 50 vol.% H_2_). It is not as expected, because hydrogen, due to its physicochemical and thermochemical properties, should show better reducing properties. Most likely, the sorptive properties of the reactants play an important role in this case. However, further studies showed better reduction results when the use of a gas mixture could be compensated for by an increased amount of reductant, as shown in Table 17.

##### Dust Reduction

Reduction conditions for (P11A) dust are summarized in Table 18. Two reduction tests were carried out at the temperature of 1000 °C, the total flow rate of hydrogen and carbon monoxide was 1.9 L/min (0.95 L/min for CO and 0.95 L/min for H_2_).

Figure 10 shows the course of reduction resulting from the amount of condensed water (ml H_2_O/5 min) from the post-reaction gases (the values shown are average values from two experiments). It can be seen that the reduction process is practically completed after 140 min, while the maximum reduction rate takes place during the first 40 min of reduction.

After the reduction process, a chemical analysis of samples was performed to test the total iron content—Fe_t_, metallic iron—Fe_met._ (A) and an instrumental analysis—X-ray diffraction patterns of (B); the results are presented in Table 19.

The mean content of total iron in a reduced sample after the A analysis was 92.3 wt.%, while the analysis results performed by the X-ray diffraction method showed an average value of 90.0 wt.%. In the sample before reduction, the total iron content was 82.8 wt.%; by considering the total reduction of oxides to metallic iron, the required weight loss is 15.43 wt.%, which should result in an increase in the total iron concentration in the sample to 97.9 wt.%. This indicates that chemical analyses are burdened with a considerable error, mainly in determining metallic iron, because the metallic iron content in the sample before reduction is 35.4 wt.%, while after reduction it increases by 2.4 wt.% only.

As a final conclusion of the results of dust reduction (P11A) with the use of a gas mixture of 50 vol.% CO + 50 vol.% H_2_, it should be stated that no unequivocal conclusions can be drawn regarding the real reduction and metallization degrees. The reason for this is most likely a high dust contamination with substances significantly affecting the results of the analysis of iron content: Fe^3+^, Fe^2+^ and Fe_met_.

##### Sludge Reduction

The conditions for the sludge reduction process (16) are summarized in Table 20. Two research tests were performed for the same flow rate of the reducer (50 vol.% CO + 50 vol.% H_2_), temperature, and reduction time. The course of the reduction was observed through condensation of water from the cooled post-reaction gas (Figure 11).

As a specific characteristic of the course of this reduction process, it was noticed that after 180 min the amount of condensable water was still quite large compared to the initial stage of the reduction process. Such a state could indicate that some unreduced iron oxides remained in the system. In order to explain this phenomenon, chemical composition analyses were performed for the contents of total iron and metallic iron—Table 21.

A significant difference in the determination of Fe_t_ in repeated tests in the case of the A laboratory can be seen and it was 60% abs. The difference in the determination results made by the B Laboratory is much smaller—the difference in the determination of Fe_t_ was 2.90% abs. Due to the occurrence of these differences, calculations were made on the percentage change in the total iron content of Fe_t_ after reduction, taking into account the oxygen loss in the reduction process. The calculations showed that the percentage of Fe_t_ after reduction of the sample mass—assuming 100% reduction, should increase to 76.00 wt.%. According to the obtained results on the Fe_t_ determination in sludge samples from S16/19-09 testing, it was indicated that the average content of Fe_t_—75.65 wt.%—is the most representative and the reduction ratio R has reached 99.5%. Taking into account the discrepancy between the reduction process and the calculated reduction ratio, an additional experiment of a sludge sample reduction was carried out with an increased reductant flow (CO + H_2_) equal to 4.8 L/min. The results of the experiment are shown in Table 22, comparing with the results of reduction with CO gas.

These results indicate that at the temperature of 1000 °C it is possible to achieve high degrees of iron-bearing sludge reduction, both with the use of pure CO as a reducing agent and a gas mixture of 50 vol.% CO + 50 vol.% H_2_.

## 4. Conclusions

The reduction studies concerned samples from three groups of iron-bearing waste materials: I. mill scale, II. dusts from foundries, III. metallurgical sludge. All samples were reduced with the use of reducing gases: (a) pure carbon monoxide, (b) gas mixtures: (50 vol.% CO + 50 vol.% H_2_) at the temperature of 1000 °C in a laboratory rotary furnace and some of the samples were reduced at the range of 850 ÷ 1000 °C. The reduction time was 180 min in all cases. 

The obtained results allow to draw the following conclusions: Research on the impact of temperature on the mill scale reduction process shows that at 900 and 1000 °C it is possible to achieve a high metallization degree of the sample equal to 99.6%. However, if carbon is present in the reaction system, the reduction process should be carried out at a temperature of 1000 °C to effectively use the carbon reduction potential.Dust reduction studies have shown a large influence of the dust particle size composition and the dust carbon content on the reduction results.The reduction rate of iron oxides from fine-particles is higher by 65% compared to the reduction rate of medium-particle dust. The average reduction rate for the dust sample with increased carbon content (3.32 wt.% C) is 36.8% faster than that of the sample with lower carbon content (0.46 wt.% C).In the case of reduction of iron oxides contained in dust with pure carbon monoxide the achieved metallization degree is equal to 66.9%.In the studies on the reduction of iron oxides from the sludge it was found that during the three-hour reduction period with carbon monoxide it was possible to achieve the reduction degree with an average level of 97.6%.When a mixture of CO and H_2_ gases was used in the reduction process, high levels of metallization were obtained in the case of sludge (95.2%) and mill scale (87.0%). However, in the case of dust reduction, no unequivocal reduction results were obtained. The reason for this is most likely the contamination of iron-bearing waste materials with technological substances used in foundries.

In the general conclusions of the research, it should be stated that it is possible to achieve high degrees of reduction and metallization of the investigated iron-bearing waste materials using gas reducers, both in the form of CO and a mixture of (50 vol.% CO + 50 vol.% H_2_) under the conditions of a rotary kiln in a temperature range of 900 ÷ 1000 °C, during the process not exceeding 180 min. The resulting product can be a valuable charge in steelmaking processes.

## Figures and Tables

**Figure 1 materials-14-01914-f001:**
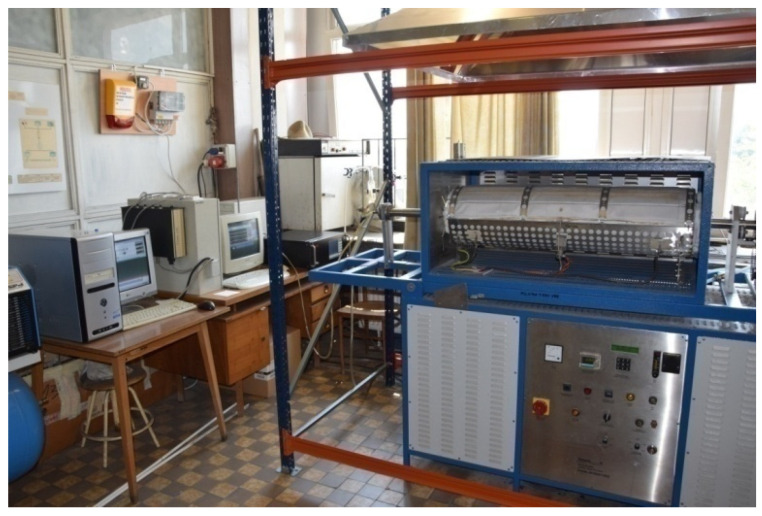
Laboratory stand for testing reductions in a rotary furnace after the upgrade [26].

**Figure 2 materials-14-01914-f002:**
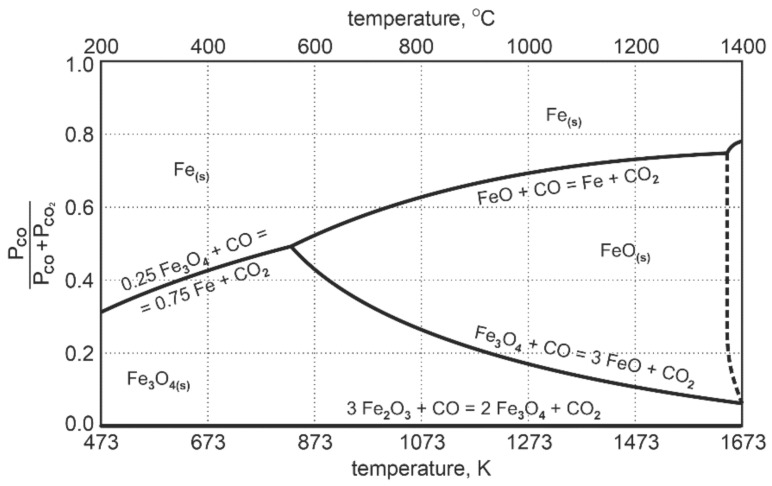
Fe-O-C equilibrium system during reduction of intermediate products of iron oxides [24].

**Figure 3 materials-14-01914-f003:**
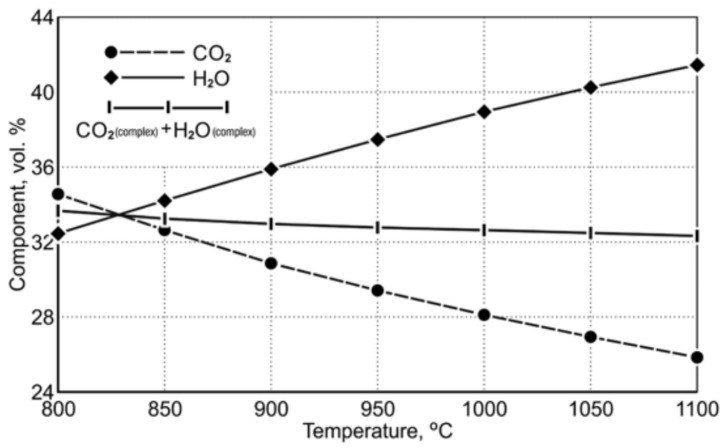
The course of equilibrium curves of (CO_2_ and H_2_O) FeO reduction reaction by using CO and H_2_ in the temperature range of 850 ÷ 1050 °C [20].

**Figure 4 materials-14-01914-f004:**
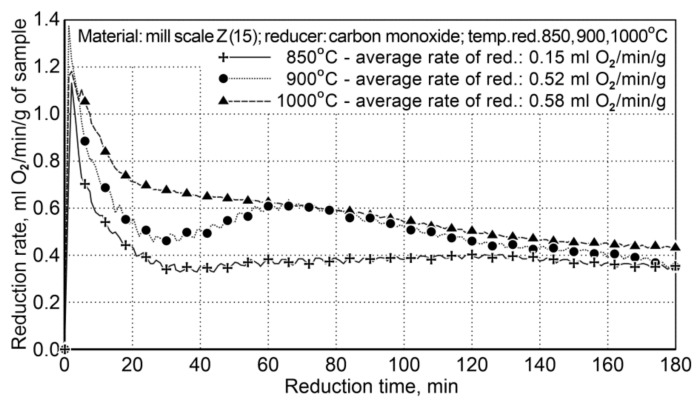
The course of the mill scale reduction (Z15) at three process temperatures: 850, 900, and 1000 °C.

**Figure 5 materials-14-01914-f005:**
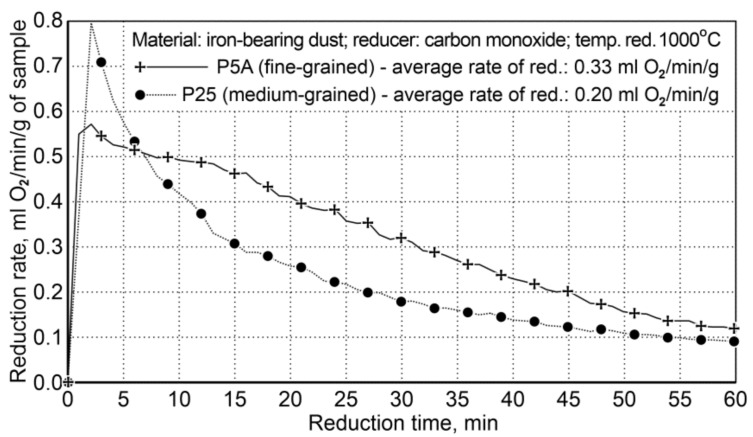
The course of reduction of iron oxides from P5A and P25 iron-bearing dust (Variant I).

**Figure 6 materials-14-01914-f006:**
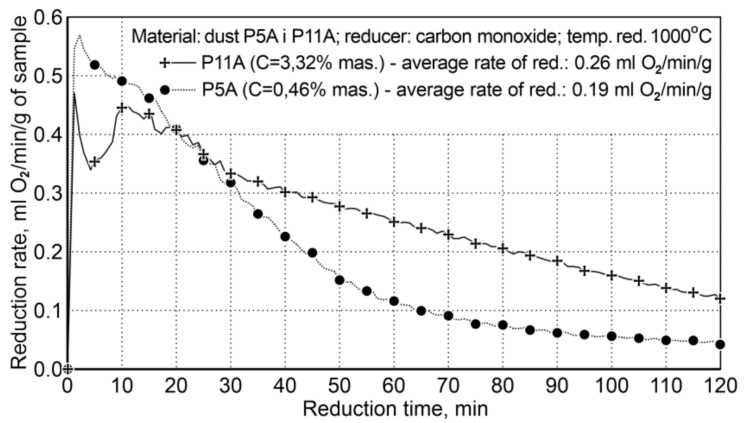
The course of reduction of iron oxides from P5A and P11 iron-bearing dust (variant II).

**Figure 7 materials-14-01914-f007:**
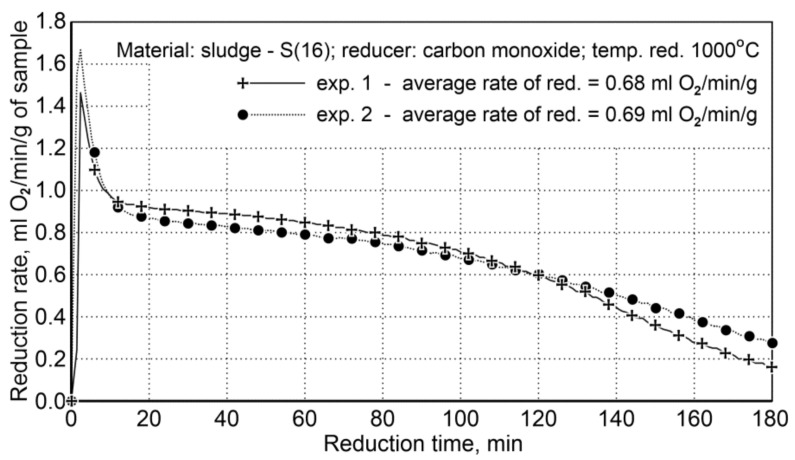
Course of reduction of sludge (S16) with carbon monoxide at 1000 °C in the FDHK 40/600/1200 rotary kiln.

**Figure 8 materials-14-01914-f008:**
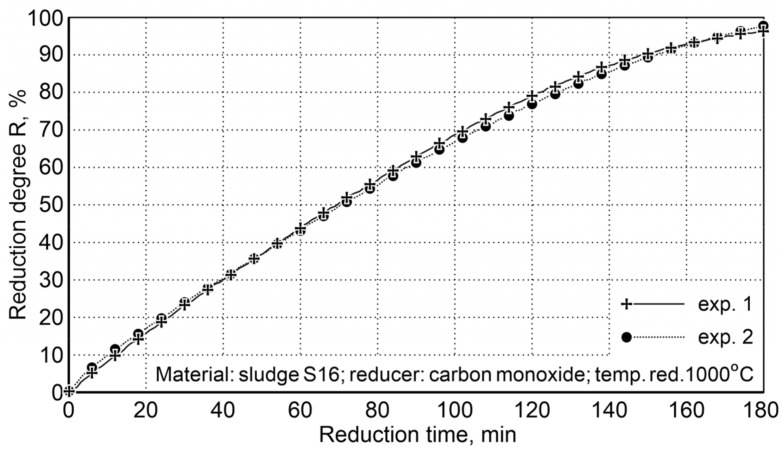
The degree of sludge reduction (S16) at 1000 °C with carbon monoxide in the laboratory rotary kiln FDHK 40/600/1200.

**Figure 9 materials-14-01914-f009:**
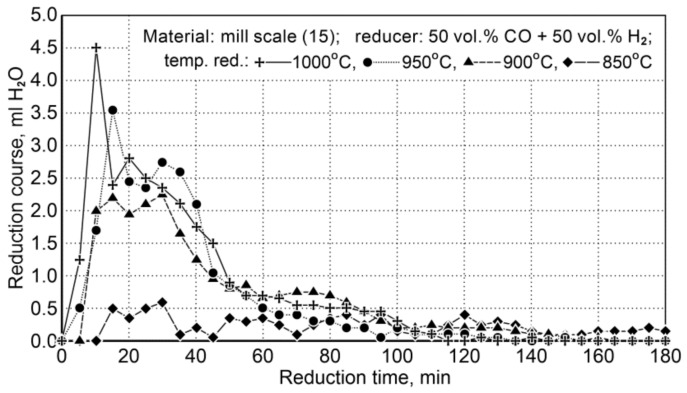
The course of the mill scale reduction (Z15) in the temperature range 850 ÷ 1000 °C using reducing gases (50 vol.% CO + 50 vol.% H_2_).

**Figure 10 materials-14-01914-f010:**
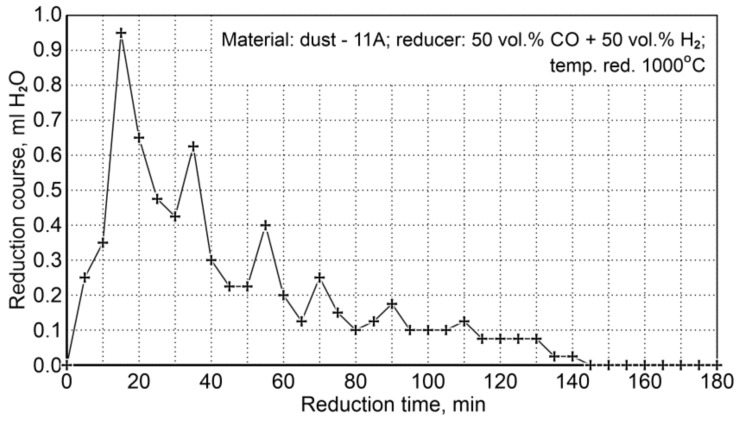
The course of P11A iron-bearing dust reduction at 1000°C with reducing gases (50 vol.% CO + 50 vol.% H_2_).

**Figure 11 materials-14-01914-f011:**
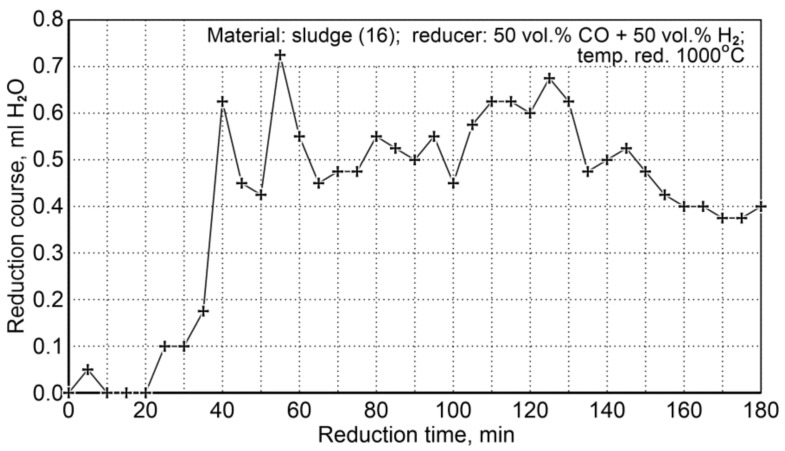
The course of sludge reduction (S16) at 1000 °C with reducing gases (50 vol.% CO + 50 vol.% H_2_).

**Table 1 materials-14-01914-t001:** The amount of waste materials received per 1 t of steel production [12].

Material Type	Quantity, kg/t of Steel
Blast furnace slag	150 ÷ 350
Steel converter slag	100 ÷ 130
Secondary metallurgy	2 ÷ 16
Steel slag from electric arc furnaces	130 ÷ 150
Scale and mill scale	10 ÷ 20
Sintering dust	10 ÷ 15
Dust from electric arc furnaces	10 ÷ 15
Blast furnace sludge	1 ÷ 2
Basic oxygen furnace sludge (fine fractions)	15 ÷ 17
Basic oxygen furnace sludge (coarser fractions)	5 ÷ 6
Blast furnace sludge	<1.0
Blast furnace slag	150 ÷ 350

**Table 2 materials-14-01914-t002:** Percentage contribution of technology-specific metallurgical operations in dust emissions [15].

Type of Technological Process	Dust Pollution, in wt.%
Storage and preparation of raw materials	~5
Ore sintering	~12
Coal coking	~20
Production of pig iron and cast iron in blast furnaces and cupolas	~20
Steel production in converters and electric furnaces	4 ÷ 10
Cooper melting	~10
Aluminum electrolysis	~10

**Table 3 materials-14-01914-t003:** Content of total iron, elemental carbon, basicity of (CaO/SiO_2_), and other components in the scale (wt.%) [20].

Sample No	Components, wt.%							
	Fe_2_O_3_	C	CaO	SiO_2_	O	Al_2_O_3_	Zn	S
3	83.67	0.14	0.80	2.20	–	0.32	–	–
6	81.67	0.27	1.44	3.42	0.50	0.76	–	–
10	72.83	0.07	0.52	5.41	1.49	0.51	–	0.1
15	85.10	0.25	0.14	2.35	–	0.38	–	–
20	85.60	0.03	0.56	1.00	–	0.25	–	–

**Table 4 materials-14-01914-t004:** Particle size distribution of scale in particular samples, wt.% [26].

Fraction, mm	Sample No				
	3	6	10	15	20
<0.5	40.80	43.46	5.76	23.49	14.41
0.5 ÷ 1.0	19.15	32.82	9.27	7.25	26.94
1.0 ÷ 2.0	9.01	9.14	9.02	5.12	10.40
2.0 ÷ 3.0	9.26	5.89	10.53	7.00	17.67
3.0 ÷ 5.0	12.14	4.88	17.67	13.67	21.80
>5.0	9.64	3.81	47.74	43.48	8.77

**Table 5 materials-14-01914-t005:** Chemical composition (wt.%) and basicity (CaO/SiO_2_) of selected dust samples [20].

Sample No	Components, wt.%								
	Fe	C	CaO	SiO_2_	MgO	Al_2_O_3_	Zn	S	CaO/SiO_2_
5A	87.77	3.32	0.14	3.78	0	0.43	0	0	0.04
5C	84.44	4.17	0.66	6.91	0.12	3.53	0	0.07	0.1
11A	86.4	0.46	0.18	5.28	0.28	1.89	0	0	0.03
19	41.67	8.77	14.1	4.21	0	1.45	18.33	0.7	3.35
22	88.01	2.91	0.1	6.05	0	2.78	0	0.13	0.02
23	68.73	0.99	0.24	27.93	0.2	1.72	0	0.18	0.01
25	69.87	3.35	0.32	12.19	0.12	12.15	0.2	0.37	0.03

**Table 6 materials-14-01914-t006:** Particle size composition of dusts in particular samples, wt.% [26].

Fraction, mm	Sample No						
	5A	5C	11A	19	22	23	25
<0.5	98.6	97.5	99.9	59.5	87.3	37	67.5
0.5 ÷ 1.0	0.8	1.8	0.1	11.8	10	56.4	13.3
1.0 ÷ 2.0	0.1	0.7	0	4.4	1.6	6.6	2
2.0 ÷ 3.0	0.1	0	0	4.6	0.5	0	1.6
3.0 ÷ 5.0	0.4	0	0	7.8	0.3	0	3.3
>5.0	0	0	0	11.9	0.3	0	12.3

**Table 7 materials-14-01914-t007:** Chemical composition of selected sludge samples, wt.% [20].

Sample No	Components, wt.%								
	Fe	C	CaO	SiO_2_	MgO	Al_2_O_3_	Zn	S	CaO/SiO_2_
14	59.87	4.92	0.84	22.18	0.28	2.89	0.5	0.27	0.04
16	62.76	1.88	1.02	19.04	0.28	1.45	0.4	0.5	0.05
32	78.17	3.36	0.18	0.58	–	0.38	–	0.1	0.31

**Table 8 materials-14-01914-t008:** Particle size of sludge, wt.% [26].

Fraction, mm	Sample No		
	14	16	32
<0.5	89.66	81.99	36.97
0.5 ÷ 1.0	7.34	9.25	56.39
1.0 ÷ 2.0	1.70	2.27	6.64
2.0 ÷ 3.0	0.80	1.80	0.00
3.0 ÷ 5.0	0.45	1.67	0.00
>5.0	0.05	3.02	0.00

**Table 9 materials-14-01914-t009:** Equilibrium gas composition for the reaction of: (1) FeO + CO =; (2) FeO + H_2_ =; (3) FeO + 0.5CO + 0.5H_2_ = in the temperature range of 850–1050 °C, vol.%. and the equilibrium share of Fe and FeO in the reduction reaction, Equation (3), wt.% [26].

Temp., °C	FeO + CO=		FeO + H_2_=		FeO + 0.5CO + 0.5H_2_=				
	CO	CO_2_	H_2_	H_2_O	CO_2_k*	H_2_Ok*	CO_2_k + H_2_Ok	Fe	FeO
850	67.45	32.55	65.83	34.17	16.25	17.08	33.33	18.61	47.91
900	69.21	30.79	64.17	35.83	15.40	17.91	33.31	18.60	47.92
950	70.66	29.34	62.58	37.42	14.67	18.71	33.38	18.64	47.86
1000	71.95	28.05	61.12	38.88	14.02	19.44	33.46	18.69	47.80
1050	73.13	26.87	59.81	40.19	13.43	20.09	33.52	18.72	47.76

CO_2_k*, H_2_Ok*—concentration of CO_2_ and H_2_O in the complex reduction reaction with CO and H_2_.

**Table 10 materials-14-01914-t010:** Conditions for reducing iron oxides for scale Z (15) and Fe_t_, Fe^3+^ and Fe^2+^ iron content in a sample for particular variants.

Variant	Designation of the Experiment	Temperature, °C	Sample Weight, g	Process Time, min	Flow Rate CO, L/min	Iron Content in the Sample, wt.%			C, wt.%
						Fe_t_	Fe^3+^	Fe^2+^	
I	Z15/01-09	850	384.63	180	1.9	75.6	24.5	48.7	0.25
II	Z15/06-09	900	380.97	180	1.8	75.6	24.5	48.7	0.25
III	Z15/18-07	1000	365.86	180	5.6	75.6	24.5	48.7	0.25
	Z15/31-08	1000	367.39	180	1.8	75.6	24.5	48.7	0.25

**Table 11 materials-14-01914-t011:** Conditions for reduction of P(5A) and P(11A) dusts and Fe_t_, Fe^3+^ and Fe^2+^ iron content in a sample.

Sample No	Designation of the Experiment	Temp., °C	Sample Weight, g	Process Time, min	Flow Rate CO, L/min	Iron Content in the Sample, wt.%		
						Fe_t_	Fe^3+^	Fe^2+^
dust (5A)	P5A/04-09	1000	265.96	120	2.2	77.5	41.9	12.0
dust (11A)	P11A/20-07	1000	446.34	120	2.1	82.8	11.2	37.2

**Table 12 materials-14-01914-t012:** Conditions for reduction of P(5A) and P(25) dust and Fe_t_, Fe^3+^ and Fe^2+^ iron content.

Sample No	Designation of the Experiment	Temp., °C	Sample Weight, g	Process Time, min	Flow Rate CO, L/min	Iron Content in the Sample, wt.%		
						Fe_t_	Fe^3+^	Fe^2+^
dust (5A)	P5A/04-09	1000	265.96	120	2.2	77.5	41.9	12.0
dust (25)	P25/26-07	1000	234.39	120	1.6	46.4	9.6	4.81

**Table 13 materials-14-01914-t013:** Conditions for reduction of S(16) sludge and the iron content of Fe_t_, Fe^3+^ and Fe^2+^ in a sample before reduction.

Experiment	Designation of the Experiment	Temp., °C	Sample Weight, g	Process Time, min	Flow Rate CO, L/min	Iron Content in the Sample, wt.%		
						Fe_t_	Fe^3+^	Fe^2+^
1	S16/07-09	1000	252.48	180	1.6	60.2	27.9	30.9
2	S16/08-09	1000	267.65	180	1.6	60.2	27.9	30.9

**Table 14 materials-14-01914-t014:** Conditions for research testing on reduction of iron oxides from waste materials in a FDHK rotary furnace by applying a mixture of reducing gases consisting of 50% of CO + 50% of H_2_.

Sample On	Temperature, °C		Fe_t_, wt.%	C, wt.%	Process Time, min	Reducer
	Non-Isothermal	Isothermal				
dust (11A)	850 ÷ 1000	1000	82.8	0.46	180	50% CO + 50% H_2_
scale (15)	850 ÷ 1000	1000	75.6	0.25	180	50% CO + 50% H_2_
scale (15)	850 ÷ 950	950	75.6	0.25	180	50% CO + 50% H_2_
scale (15)	850 ÷ 900	900	75.6	0.25	180	50% CO + 50% H_2_
scale (15)	–	850	75.6	0.25	180	50% CO + 50% H_2_
sludge (16)	850 ÷ 1000	1000	60.2	1.88	180	50% CO + 50% H_2_

**Table 15 materials-14-01914-t015:** Conditions for reduction of scale (15) and the iron content of Fe_t_, Fe^3+^ and Fe^2+^ in a sample during the reduction process by applying a mixture of reducing gases consisting of 50 vol.% of the CO + 50 vol.% of H_2._

Designation of the Experiment	Temp., °C	Sample Weight, g	Process Time, min	Reducing Gas Flow, L/min	Iron Content in the Sample, wt.%			
					Fe_t_	Fe^3+^	Fe^2+^	Fe_met._
Z15/26-09	1000	380.98	180	2.8	75.6	24.5	48.7	2.4
Z15/29-09	950	380.63	180	2.8	75.6	24.5	48.7	2.4
Z15/28-09	900	382.93	180	2.8	75.6	24.5	48.7	2.4
Z15/25-09	850	378.90	180	2.8	75.6	24.5	48.7	2.4

**Table 16 materials-14-01914-t016:** Mill scale (Z15) reduction results by using carbon monoxide and hydrogen (50 vol.% CO + 50 vol.% H_2_) and pure carbon monoxide at 1000 °C.

Reducer	Reducer Flow Rate L/min	Fe_t_, wt.%	Fe_met._, wt.%	(Fe_met._/Fe_t_)·100%
100% vol. CO	1.8	83.4	40.8	48.9
50 vol.% CO + 50 vol.% H_2_	2.8	85.3	33.7	39.5

**Table 17 materials-14-01914-t017:** Mill scale (Z15) reduction results by using carbon monoxide and hydrogen (50 vol.% CO + 50 vol.% H_2_) and pure carbon monoxide at 1000 °C with an increased amount of reducer gas.

Reducer	Reducer Flow Rate L/min	Fe_t_, wt.%	Fe_met._, wt.%	(Fe_met._/Fe_t_)·100%
100 vol.%. CO	4.8	83.2	82.9	99.6
50 vol.% CO + 50 vol.% H_2_	4.8	80.2	65.5	81.7
50 vol.% CO + 50 vol.% H_2_	6.0	82.3	71.6	87.0

**Table 18 materials-14-01914-t018:** Reduction conditions for P (11A) dust and iron content of Fe_t_, Fe^3+^ and Fe^2+^ in a sample in the reduction process by applying a reducing gas mixture consisting of 50 vol.% of CO + 50 vol.% of H_2._

Designation of the Experiment	Temp., °C	Sample Weight, g	Process Time, min	Reducing Gas Flow, L/min	Iron Content in the Sample, wt.%			
					Fe_t_	Fe^3+^	Fe^2+^	Fe_met._
P11A/21-09	1000	442.97	180	1.9	82.8	11.2	37.2	35.4
P11A/22-09	1000	441.63	180	1.9	82.8	11.2	37.2	35.4

**Table 19 materials-14-01914-t019:** Chemical analysis results of samples with P11A dust after reduction at 1000 °C with carbon monoxide (1) and hydrogen (50 vol.% CO + 50 vol.% H_2_) (2).

Laboratory	Experiment P11A/21-09 (1)			Experiment P11A/22-09 (2)		
	Fe_t_, wt.%	Fe_met,_ wt.%	(Fe_met./_Fe_t_)˖100%	Fe_t_, wt.%	Fe_met._, wt.%	(Fe_met./_Fe_t_)˖100%
A	93.2	37.9	40.7	91.3	37.8	41.4
B	90.3	–	–	89.7	–	–

**Table 20 materials-14-01914-t020:** Conditions for sludge reduction (16) and the iron content of Fe_t_, Fe^3+^ and Fe^2+^ in a sample during reduction process in rotary furnace by using a reducing gas mixture containing of (50 vol.% CO + 50 vol.% H_2_).

Designation of the Experiment	Temp., °C	Sample Weight, g	Process Time, min	Reducing Gas Flow, L/min	Iron Content in the Sample, wt.%			
					Fe_t_	Fe^3+^	Fe^2+^	Fe_met._
S16/19-09	1000	265.82	180	0.8	60.2	27.9	30.9	1.4
S16/20-09	1000	261.19	180	0.8	60.2	27.9	30.9	1.4

**Table 21 materials-14-01914-t021:** The content of total iron of Fe_t_ and Fe_met._ in samples of sludge (S16) after reduction at 1000°C by using carbon monoxide and hydrogen (50 vol.% of CO + 50 vol.% of H_2_).

Laboratory	Experiment S16/19-09			Experiment S16/20-09		
	Fe_t_, wt.%	Fe_met,_ wt.%_._	(Fe_met./_Fe_t_) + 100%	Fe_t_, wt.%	Fe_met._, wt.%	(Fe_met./_Fe_t_) + 100%
A	75.45	38.25	50.7	68.85	33.39	49.2
B	75.85	–	–	72.95	–	–
average	75.65	–	–	70.90	–	–

**Table 22 materials-14-01914-t022:** The results of the reduction of S(16) sludge at 1000 °C using carbon monoxide as a reducing agent and a mixture (50 vol.% CO + 50 vol.% H_2_).

Reducer	Reducer Flow Rate L/min	Fe_t,_ wt.%	Fe_met._, wt.%	(Fe_met._/Fe_t_)·100%	Reduction Ratio R
100 vol.%. CO	1.6	–	–	–	97.6 *
50 vol.% CO + 50 vol.% H_2_	4.8	78.8	75.0	95.2	99.5

* this value is the calculated degree of reduction R.

## Data Availability

The study did not report any data.

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
