# Peer review of "Research on Reduction of Selected Iron-Bearing Waste Materials"

_materials, 2021, doi:10.3390/ma14081914_

Round 1
Reviewer 1 Report
The article develops a research focused on steel production, as well as experiments with a technique to reduce waste materials in steel production and obtain improvements in steel production, although in the abstract no these improvements appear clearly so the wording of the abstract in this regard needs to be improved. The research is interesting and presents contributions in this field, but the article has shortcomings to improve in the paper review process.
1- Which units are Mg and kg / Mg?
2- Why is there the numerical denomination of Samples in Tables 3, 4: sample 3,6,10, 15 ??
3- How the compositions and fractions of each sample are determined in Tables 5, 6, 7 ...
4- Describe what type of reaction represents equation 1 and why
5- Table 10, appears with deficiencies in the titles or heading of the Table
6- What differences and advantages / disadvantages do metallic iron (Femet) represent compared to other types of iron? I don’t understand the distinction
7- Could you indicate which statistical variables and errors have been calculated / measured in the present research?
8- Many acronyms remain unexplained or clarified as they may be, it is not known what they are and have not been properly introduced: VAD, VOD, LF, OA Taguchi, ABB Uras, EAF
9- The conclusions do not understand why a high level of metallization is required, please clarify this part
10- In the article Neither the objectives nor the conclusions are clear, and the abstract should also be improved, clarifying which are the most remarkable milestones of the present manuscript.
11- What contributions does the present research represent?
12- In what areas, aspects, industries and companies does your research bring benefits?
Author Response
Comments and Suggestions for Authors:
The article develops a research focused on steel production, as well as experiments with a technique to reduce waste materials in steel production and obtain improvements in steel production, although in the abstract no these improvements appear clearly so the wording of the abstract in this regard needs to be improved. The research is interesting and presents contributions in this field, but the article has shortcomings to improve in the paper review process.
1- Which units are Mg and kg / Mg?
A correction was introduced as suggested by the Reviewer.
2- Why is there the numerical denomination of Samples in Tables 3, 4: sample 3,6,10, 15 ??
From the waste materials (scale) obtained from the steelworks and metallurgical plants, the samples listed in the table met the condition above 40% of the total iron content. These were samples 3, 6, 10, 15, 20.
3- How the compositions and fractions of each sample are determined in Tables 5, 6, 7 ...
As in the case of scale, dust and sludge, the samples obtained from production plants were certified, and on this basis the chemical composition of selected of materials tables were prepared. The obtained individual waste materials were prepared for testing by fractionating into individual grain classes.
4- Describe what type of reaction represents equation 1 and why
Reaction (1) is a reduction reaction. According to the calculation of the reaction effect, the energy demand for this reaction is ~ + 13 kJ / mol.
5- Table 10, appears with deficiencies in the titles or heading of the Table
It is an editor error. This has been corrected.
6- What differences and advantages / disadvantages do metallic iron (Femet) represent compared to other types of iron? I don’t understand the distinction
The amount of metallic iron was converted from the amount of total iron. In the reactions taking place in metallurgical aggregates, the quantity converted into metallic iron is important - we are talking here about the richness of the charge. Metallic iron does not require reduction. Iron oxides require ego-reducing material and energy to carry out the reduction reaction. This is an additional cost.
7- Could you indicate which statistical variables and errors have been calculated / measured in the present research?
No statistical studies were conducted in this part of the study. This was the subject of detailed research during the PhD thesis of one of the co-authors.
8- Many acronyms remain unexplained or clarified as they may be, it is not known what they are and have not been properly introduced: VAD, VOD, LF, OA Taguchi, ABB Uras, EAFdo
VAD, VOD, LF, EAF – as suggested by the Reviewer, they were included in the text;
OA Taguchi - proper name. This method was used at the stage of designing the research to evaluate the course of physicochemical phenomena. This made it possible to determine which of the factors have the greatest impact on the studied process and indicate their categorization and grading;
ABB Uras 14 – w tekÅ›cie na str. 10 – „… the ABB Uras 14 gas analyzer” – this is a equipment, the gas analyzer.
9- The conclusions do not understand why a high level of metallization is required, please clarify this part
As mentioned before, it is about the suitability of metallic iron as a feedstock for metallurgical processes - which can be used directly. In the case of iron oxides, a reductor and energy are needed to carry out the reduction reaction, which generates additional costs.
10- In the article Neither the objectives nor the conclusions are clear, and the abstract should also be improved, clarifying which are the most remarkable milestones of the present manuscript.
As suggested by Reviewer, this has been corrected in the text.
11- What contributions does the present research represent?
As suggested by Reviewer, this has been corrected in the text. Use of waste materials as input materials (consistent with point 9).
12- In what areas, aspects, industries and companies does your research bring benefits?
Thus, we contribute to sustainable development and environmental protection consistent with the environmental management strategy.
Best regards and great thanks for your contribution to improving the quality of the article
Authors

Reviewer 2 Report
Dear Authors,
your paper "Research on Reduction of Selected Iron-Bearing Waste Materials" investigates the possibility to recover iron from iron-bearing wastes by the use of rotary oven feed by reduction atmospheres (CO and CO/H2 50:50)
The paper is interesting since it discusses a hot topic. However the main flaw of your work is hot the text is organized.
First of all, I suggest to divide into sub-paragraph the introduction. Please, split the steelmaking data from the analysis of the wastes and from the hydrogen route. In addition, I found three papers belonging to the same project related on the recovery of steelmaking fines wtih the steel-shop itself. I think some aspect can be included in the introduction as well some results on the thermal characterization of the briquettes can be used for the discussion of results.
1) Investigation on the chemical and thermal behavior of recycling agglomerates from EAF by-products
Willms, T., Echterhof, T., Steinlechner, S., ...Mapelli, C., Preiss, S.
Applied Sciences (Switzerland), 2020, 10(22), pp. 1–14, 8309
2) Fabrication of agglomerates from secondary raw materials reinforced with paper fibres by stamp pressing process
Echterhof, T., Willms, T., Preiss, S., ...Steinlechner, S., Unamuno, I.
Applied Sciences (Switzerland), 2019, 9(19), 3946
3) Developing a new process to agglomerate secondary raw material fines for recycling in the electric arc furnace - The fines2EAF project
Echterhof, T., Willms, T., Preiß, S., ...Mudersbach, D., Griessacher, T.
Metallurgia Italiana, 2019, 111(5), pp. 31–40.
Then, separate the experimental procedure from the results, because currently is really hard to understand the results. I also suggest to merge results and discussion, and separate the discussion from the conclusions. In the experimental procedure, please indicate only the methodologies used and the characteristics of the charging materials and so on (Table 10 to Table 15, Table 17, Table 19).
In addition, please find a list of specific comments/remarks/corrections to improve the quality of your paper:
a) abstract: "Only in the scale sample, the level of metalliza-tion achieved the level required when the material is used as a metallic charge in electric steel mills". Why? the sludges reached a metalizzation rate higher than scale. Please clarify
b) Introduction: please use "t" instead of "Mg" along the whole manuscript
c) Introduction, p.2: "The largest amount are slags: blast furnace slag, steel slag, basic oxygen furnace slag from continuous casting of steel and electric arc furnaces". This part of the sentence is not clear. What the Authors would mean by "slag from continuos casting and electric arc furnace"? Please, divide the different source of iron-bearing waste into BF-BOF route and EAF route.
d) introduction, p.3: please replace "carb" with "carbon"
e) introduction, p.3: please correct "vistite" with "wustite"
f) introduction, p.3: paragraph on hydrogen: This paragraph is not completely link with the previous. Please, separate by sub-paragraph and enrich it with more detail, especially regarding temperature, costs and processes
g) sampling, p.4: "Grain analysis of the samples used in the subsequent stages of the research was also performed". Please, specify procedure, testing machine and so on.
h) paragraph 2.1.1., p.4. "The summary of scale sample grain analysis results is presented in Table 4. It can be notices that the two smallest fractions of < 0.5 mm and 0.5 – 1.0 mm constitute the largest share, as well as, the fraction with the largest grain of 5.0 mm (three samples)." Have the Authors investigated if the chemical composition differs from fraction to fraction? This aspect would be important if a sizing is performed
i) 2.21. devices used for testing, p.6: "Alundum retort capped at the bottom had a hermetic closure at the top, which was equipped with nozzles in order to reduce gas supply and capture the post-reaction gases (Fig. 1)." I suppose is a typo
j) 2.2.1. devices used for testing, p.6: "ribbing in the form of longitudinal slats - blades mounted inside the retort." I suppose it is a typo
h) 2.3. Tests on indirect reductions carried out in stationary conditions, p.7: From the Bauer-Glaessner diagram, at 1000 °C an atmosfere containing the 70% of CO is required. Why the Authors stated 30%? Please clarify.
l) 2.3. Tests on indirect reductions carried out in stationary conditions, p.8: "This indicates reducer wear of approx. 70%. The process of reduction to metallic iron reached the level of 30%. In order to achieve full reduction of FeO to metallic iron, the reducing agent in the system should exceed the amount of: n=100/%CO2 (2)
where, n – excessive agglomeration of CO required for complete reduction of FeO to me-tallic iron, %CO2 – equilibrium concentration of CO2 in the system at a given temperature, expressed in percent." This part is not completely clear. The reation CO/FeO for a stoichometric reduction to iron is equal to 0.39 while the required amount of CO is the 28% of the whole amount of reagent (FeO+CO). Moreover, the metallization ratio is 0.78. What the Authors would mean is that at 1000 K only 28% CO is used to fulfil reduction even the equilibrium requires at least an atmosphere with the 70% of CO? Please, try to be more clear in this point.
m) paragraph 2.4, p.9: "CO2". I suppose is CO
n) paragraph 2.4.1, p.9: "The tests on reduction of iron-bearing waste material samples were carried out in the stationary layer in the Tamman furnace. Results of this research were presented at the 2017 AISTECH International Conference and 2019 Iron and Steelmaking International conference [19,22]." Please, summarize here the main results for who has no access to the AISTECH proceedings
o) p.10 below Table 10.: "O2", please the 2 as subscript
Best regards
Author Response
Responses to the Reviewer's comments and recommendations
Dear Authors,
your paper "Research on Reduction of Selected Iron-Bearing Waste Materials" investigates the possibility to recover iron from iron-bearing wastes by the use of rotary oven feed by reduction atmospheres (CO and CO/H2 50:50)
The paper is interesting since it discusses a hot topic. However the main flaw of your work is hot the text is organized.
First of all, I suggest to divide into sub-paragraph the introduction. Please, split the steelmaking data from the analysis of the wastes and from the hydrogen route. In addition, I found three papers belonging to the same project related on the recovery of steelmaking fines wtih the steel-shop itself. I think some aspect can be included in the introduction as well some results on the thermal characterization of the briquettes can be used for the discussion of results.
In the chapter „1. Introduction” two sections were introduced „1.1. Quantitative characteristics of metallurgical waste” and „1.2. Qualitative characteristics of metallurgical waste”
1) Investigation on the chemical and thermal behavior of recycling agglomerates from EAF by-products
Willms, T., Echterhof, T., Steinlechner, S., ...Mapelli, C., Preiss, S. Applied Sciences (Switzerland), 2020, 10(22), pp. 1–14, 8309
2) Fabrication of agglomerates from secondary raw materials reinforced with paper fibres by stamp pressing process
Echterhof, T., Willms, T., Preiss, S., ...Steinlechner, S., Unamuno, I.
Applied Sciences (Switzerland), 2019, 9(19), 3946
3) Developing a new process to agglomerate secondary raw material fines for recycling in the electric arc furnace - The fines2EAF Project
Echterhof, T., Willms, T., Preiß, S., ...Mudersbach, D., Griessacher, T.
Metallurgia Italiana, 2019, 111(5), pp. 31–40.
Thank you for pointing out these newest and very interesting materials, they were used in the introduction to this article.
Then, separate the experimental procedure from the results, because currently is really hard to understand the results. I also suggest to merge results and discussion, and separate the discussion from the conclusions. In the experimental procedure, please indicate only the methodologies used and the characteristics of the charging materials and so on (Table 10 to Table 15, Table 17, Table 19).
The changes proposed by the Reviewer have been made
In addition, please find a list of specific comments/remarks/corrections to improve the quality of your paper:
- abstract: "Only in the scale sample, the level of metalliza-tion achieved the level required when the material is used as a metallic charge in electric steel mills". Why? the sludges reached a metalizzation rate higher than scale. Please clarify
Indeed, instead of: "Only in the scale sample ..." it should be “These results indicate that in the case of sludge and scale, the degree of metallization meets the requirements for charge materials used in both BOF and EAF steelmaking processes, while in the case of reduced dust, this material can be used as enriched charge in the blast furnace process”.
- Introduction: please use "t" instead of "Mg" along the whole manuscript
A correction was introduced as suggested by the Reviewer.
- Introduction, p.2: "The largest amount are slags: blast furnace slag, steel slag, basic oxygen furnace slag from continuous casting of steel and electric arc furnaces". This part of the sentence is not clear. What the Authors would mean by "slag from continuos casting and electric arc furnace"? Please, divide the different source of iron-bearing waste into BF-BOF route and EAF route.
It may be assumed that lubricants used in continuom casting and foaming slags and carbide slags from EAF route can be neglected as to the quantitative importance of these slags and only slags from BF-BOF route is considered. Moreover, these slags are not included in the research, therefore a correction has been made to the relevant expression by deleting words: „slag from continuous casting of steel and electric arc furnaces”.
- introduction, p.3: please replace "carb" with "carbon"
The wrong „carb” expression is replaced by „carbon” one
- introduction, p.3: please correct "vistite" with "wustite"
The wrong „vistite” expression is replaced by „wustite” one
- introduction, p.3: paragraph on hydrogen: This paragraph is not completely link with the previous. Please, separate by sub-paragraph and enrich it with more detail, especially regarding temperature, costs and processes
A separate paragraph was introduced to justify the choice of hydrogen as an additional reducing agent, along with the specifications of references, especially with the most recent item No 20.
- sampling, p.4: "Grain analysis of the samples used in the subsequent stages of the research was also performed". Please, specify procedure, testing machine and so on.
The grain composition analysis was carried out on woven screens with square mesh sizes: 0.5 mm; 1.0 mm; 2.0 mm; 3.0 mm and 5.0 mm. The dry material was screened on a vibrating mechanical device for 5 minutes (vibrations 300/min), after which particular fraction were weighed. A control sieving was then performed for 1 minut.
- paragraph 2.1.1., p.4. "The summary of scale sample grain analysis results is presented in Table 4. It can be notices that the two smallest fractions of < 0.5 mm and 0.5 – 1.0 mm constitute the largest share, as well as, the fraction with the largest grain of 5.0 mm (three samples)." Have the Authors investigated if the chemical composition differs from fraction to fraction? This aspect would be important if a sizing is performed
The dependence of the amount of Fe on the size of the grain fraction was not investigated, because the research program did not include their performance. The average iron content in the entire sample was used, taking into account the industrial conditions of the possible use of the test results.
- 21. devices used for testing, p.6: "Alundum retort capped at the bottom had a hermetic closure at the top, which was equipped with nozzles in order to reduce gas supply and capture the post-reaction gases (Fig. 1)." I suppose is a typo
Part of the sentence has been replaced with the phrase: “tubes for supplying reducing gas and receiving post-reaction gases”.
- 2.1. devices used for testing, p.6: "ribbing in the form of longitudinal slats - blades mounted inside the retort." I suppose it is a typo
Unfortunately, this remark is not clear to us.
- h) 2.3. Tests on indirect reductions carried out in stationary conditions, p.7: From the Bauer-Glaessner diagram, at 1000 °C an atmosfere containing the 70% of CO is required. Why the Authors stated 30%? Please clarify.
In the CO-CO2 atmosphere at the temperature of 1000 C, FeO is reduced in the concentration range of 70-100% CO. Thus, in the reduction process of FeO + CO = Fe + CO2 only 30% CO is involved and only 30% FeO will be reduced. This is what the text means: "only about 30% of CO is used for the reduction reaction of FeO to Femet. After that, the system reaches a state of equilibrium”
- 3. Tests on indirect reductions carried out in stationary conditions, p.8: "This indicates reducer wear of approx. 70%. The process of reduction to metallic iron reached the level of 30%. In order to achieve full reduction of FeO to metallic iron, the reducing agent in the system should exceed the amount of: n=100/%CO2 (2)
where, n – excessive agglomeration of CO required for complete reduction of FeO to me-tallic iron, %CO2 – equilibrium concentration of CO2 in the system at a given temperature, expressed in percent." This part is not completely clear. The reation CO/FeO for a stoichometric reduction to iron is equal to 0.39 while the required amount of CO is the 28% of the whole amount of reagent (FeO+CO). Moreover, the metallization ratio is 0.78. What the Authors would mean is that at 1000 K only 28% CO is used to fulfil reduction even the equilibrium requires at least an atmosphere with the 70% of CO? Please, try to be more clear in this point.
As already mentioned, the equilibrium state of the FeO + CO = Fe + CO2 reaction makes it impossible to fully use the reduction properties of CO and reduce FeO to 100%. Thus, to achieve 100% FeO reduction, excess CO must be provided in the reaction system; this excess is calculated from the equation (2). For the reduction at 1000 C, this excess amounts to 3.33 (100/30). Thus, in the reaction system, there should be 3.33 moles of CO per mole of FeO to ensure 100% reduction of FeO.
Unfortunately, I didn't find the next expression in the article: „at 1000 K only 28% CO is used to fulfil reduction…” and I can't comment on that.
- paragraph 2.4, p.9: "CO2". I suppose is CO
Indeed, this is a mistake. Now, section 2.4 has changed to section 3 with the title "Research work and discussion" and this way the error has ceased to exist.
- paragraph 2.4.1, p.9: "The tests on reduction of iron-bearing waste material samples were carried out in the stationary layer in the Tamman furnace. Results of this research were presented at the 2017 AISTECH International Conference and 2019 Iron and Steelmaking International conference [19,22]." Please, summarize here the main results for who has no access to the AISTECH proceedings.
As suggested by the Reviewer, point 2.4.1 (currently: 3.1) has been supplemented with conclusions resulting from the research, published in the paper [21].
- 10 below Table 10.: "O2", please the 2 as subscript
Indeed, an obvious mistake. The amendment was made.
Best regards and great thanks for your contribution to improving the quality of the article
Authors

Round 2
Reviewer 1 Report
The "Research on Reduction of Selected Iron-Bearing Waste Materials" presents the investigation on reduction of selected iron-bearing waste materials. Iron-bearing waste materials in the form of dust, scale and sludge were obtained from several Polish metallurgical plants as a research material. The described method tries to reduce the waters from the metallurgic process. All the comments have been fixed and amended, so the version is ready for publication.
Author Response
Thank you for your insightful review
Reviewer 2 Report
Dear Authors,
thank you for submitting your reviewed version.
Part of the reviewers' comments were addressed but the paper still has serious flaws, mainly due to the complicated organization of the results and the absence of organic and in-depth discussion of such results.
In particular, is it not clear why the Authors tested several conditions, both with pure CO and CO+H2 atmospheres, but the comparison were done only for mill scale, while the results for dusts and sludges were not compared across the two reductants.
In additon, the presentation of the results must be improved. For instance, in the table 16-18-20 reduction rate and metallization degree must be included, because only stating them in the text make the comprehension hard.
Moreover, the results of the experiments listed in table 10-11-12-13 must be presented in form of tables. Only citing the results in the text make the comprehension hard.
Finally, a curiosity: since the experiments are dinamic test, why no graphs are shown dipicting the evolution of the system in terms of CO/CO2 measurements as well as H2O determination (only for H2+CO tests)?
Then, please find a list of some specific corrections to apply to your manuscript:
a) abstrac: please, use BF instead of BOF because it can be confused with Basic Oxygen Furnace
b) introduction, paragraph 1.1: maybe this paragraph is "qualitative" and the next is "quantitative", since the 1.2. depicts amounts, chemical compositions, and so on. Please check and correct accordingly
c) p. 3, "Dust and sludge from sinter plants or blast furnace departments indicate a high iron content ranging from 20÷45 %wt. or higher. When assessing the usability and quality of....". please add references to support your data
d) p. 3, "Sludge generated during iron and steel production processes are classified according to the following groups: pure iron-bearing sludge (with iron content exceeding 60 % wt.)...". Please, add a reference to support your classification
e) p. 4, "The assumed iron content limit in waste materials was 40 %wt. of Fe". This sentence is clear for me, but honestly can be misundarstood by other. I would propose: "the assumed iron content limit in the waste materials for a practical utilization was fixed at >40wt.% of Fe.
f) p. 4, paragraph 2.1.1. before listing the different samples' properties, it would be better to declare how the chemical composition and the granulometry were measured (machine, producer, methodology, ecc.)
g) p. 6, "Alundum retort capped at the bottom had a hermetic closure at the top, which was equipped with tubes for supplying reducing gas and receiving post-reaction gases (Fig. 1).". alumina reaction retort and alundum retort are the same device or two different devices? Because from figure 1 I can see only one furnace. Please check and if this is a repetition, adjust the text properly
h) p. 7, "Ribbing in the form of longitudinal slats - blades mounted inside the retort". I think this is a repetition of the sentence above. Please check and correct accordingly
i) p. 11, paragraph 3.3. "These research tests were performed with the use of two types of gas reducers: pure carbon monoxide and a gas mixture of- 50% CO + 50% H2. When using carbon monoxide, the reduction process was determined continuously, using automated analysis of CO and CO2 in the ABB Uras 14 gas analyzer. Thus, when using the mixture of CO and H2, the
course of reduction was observed and the amount of water condensing from the post-reaction gases was periodically recorded. After the reduction process, all samples weresubject to chemical analysis for the content of total iron (Fec) and metallic iron (Femet.). Furthermore, on selected samples, a structural analysis was performed for the content of metallic iron and other iron phases and compounds." This part should be moved in the methodology paragraph, because it describes a procedure and not a result
j) p.13, "Investigations on reduction of iron oxides from iron-bearing waste materials with the". It would be better to divide the results section in two sub-paragraph. From here the paragraph related to CO+H2 reduction starts.
k) p.14, "During tests the course of reduction was observed and recorded by the amount of condensed water from post-reaction gases. Then, the reduced samples were subject to chemical analysis for the content of total and metallic irons and to structural analysis." already declared. Please, remove it
l) p. 14, "by using as a reducing agent a mixture of reducing gases consisting
of 50% vol. of CO and "+ 50% vol. of H2". already declared. Remove it
m) p. 14, table 16. why only two conditions were compared, if the Authors did several tests both in pure CO and CO+H2?
n) p. 15, "The difference in degrees of metallization of the reduced scale, depending on the reducer being applied was 9.4% in favor for the pure carbon monoxide used". Why? An explanation is mandatory
o) p. 15, table 18. why there is not a comparison with pure CO reduction tests?
p) p. 16, paragraph 3.3.1. What is done in the tamman refers to the proceeding presented in AISTECH conference. This part must be integrated in the paragraph 3.1.
q) p. 16, paragraph 3.3.2.
why the discussion did not take into account the test performed in CO+H2?
In addition, there is no explanation the different metallization ratios and reduction rates obtained in the different samples
r) p.17, Table 21. These values should be put in the results section and not in the conclusions. Reduced samples in which atmosphere? CO or CO+H2?
Grammar errors
a) abstract: "These results indicate that in the case of sludge and scale, the degree of metallization meets the requirements for charge materials used in both blast furnace (BOF) and electric arc furnace (EAF) steelmaking processes, while in the case of reduced dust, this material can be used as enriched charge in the blast furnace process. Both pure carbon monoxide and a mixture of CO and H2 (50 vol.% CO + 50 vol.% H2) were used as a
reducer in the conducted research." please, check the font size because is different from the rest of the abstract
b) p. 2, "Metallurgical waste materials can be an alternative to conventional raw materials of construction". Replace of with for
c) p. 2, "by far exceeding other efec ts such as increased pressing force". effects
d) p. 10, "out are as follows: The". after : start with t
e) p. 11, "grain size, the basicity CaO/SiO2,". 2 as subscript
f) p.11, "Research tests on reduction of iron oxides in iron-bearing waste materials with the usage of carbon monoxide were carried out for all three groups of materials: dust, mill, scale and sludge." This sentence is redundant. Please delete
g) p. 17, "larger amount of redundant". Please correct redundant as reductant.
Author Response
Dear Authors,
thank you for submitting your reviewed version.
Part of the reviewers' comments were addressed but the paper still has serious flaws, mainly due to the complicated organization of the results and the absence of organic and in-depth discussion of such results.
In particular, is it not clear why the Authors tested several conditions, both with pure CO and CO+H2 atmospheres, but the comparison were done only for mill scale, while the results for dusts and sludges were not compared across the two reductants.
Comparative tests were also carried out for sludge samples, while in the case of dust there were difficulties, which are described in the article on page 19.
In additon, the presentation of the results must be improved. For instance, in the table 16-18-20 reduction rate and metallization degree must be included, because only stating them in the text make the comprehension hard.
The tables were supplemented with data on the metallization of samples.
Moreover, the results of the experiments listed in table 10-11-12-13 must be presented in form of tables. Only citing the results in the text make the comprehension hard.
The results of the reduction studies are also shown graphically, which should make it much easier for the reader to perceive the content.
Finally, a curiosity: since the experiments are dinamic test, why no graphs are shown dipicting the evolution of the system in terms of CO/CO2 measurements as well as H2O determination (only for H2+CO tests)?
I fully share the Reviewer's comment, now the Figures are placed, showing the course of reduction of the tested samples .Then, please find a list of some specific corrections to apply to your manuscript:
- a) abstract: please, use BF instead of BOF because it can be confused with Basic Oxygen Furnace
The correction was made.
- b) introduction, paragraph 1.1: maybe this paragraph is "qualitative" and the next is "quantitative", since the 1.2. depicts amounts, chemical compositions, and so on. Please check and correct accordingly
I believe that in this case nothing should be changed, because point 1.1 refers clearly to quantitative expressions, i.e. the amount of tonnes of waste produced, while point 1.2 refers to the properties of the waste, its chemical composition (citation: “Physicochemical properties of slags differ significantly from properties of raw materials used for their production. Slag properties depend on the type of their origin process, the quality of the input materials…..”Sludge generated during iron and steel production processes are classified according to the following groups: pure iron-bearing sludge (with iron content exceeding 60 wt.%) - sludge from wet treatment method used for exhaust gases from converter steelworks and wet sludge from scraper troughs of sinter plants, contaminated iron-bearing sludge (with iron content of 24÷56 wt.%) – sludge from blast furnace gas treatment and other sludge, such as: from neutralization processes of chemicals or oily mill scale sludge”, etc.
- c) p. 3, "Dust and sludge from sinter plants or blast furnace departments indicate a high iron content ranging from 20÷45 %wt. or higher. When assessing the usability and quality of....". please add references to support your data
Data on the characteristics of metallurgical waste can be found in the book: Jan Mróz: Recykling i utylizacja materiaÅ‚ów odpadowych w agregatach metalurgicznych (“Recycling and utilization of waste materials in metallurgical aggregates”), CzÄ™stochowa 2006.
- d) p. 3, "Sludge generated during iron and steel production processes are classified according to the following groups: pure iron-bearing sludge (with iron content exceeding 60 % wt.)...". Please, add a reference to support your classification
In Poland, for technological purposes, the following division of sludges generated in the production of iron and steel was adopted: a). pure iron-bearing sludges with an iron content above 60 wt.%, b) contaminated iron-bearing sludges with a share of 24-56 wt.% Fe and c) oily scales. This division is not formally standardized.
- e) p. 4, "The assumed iron content limit in waste materials was 40 %wt. of Fe". This sentence is clear for me, but honestly can be misundarstood by other. I would propose: "the assumed iron content limit in the waste materials for a practical utilization was fixed at >40wt.% of Fe.
The proposal was accepted. Thank you.
- f) p. 4, paragraph 2.1.1. before listing the different samples' properties, it would be better to declare how the chemical composition and the granulometry were measured (machine, producer, methodology, ecc.)
The following additions have been added:
Chemical analysis of the obtained scale, dust and sludge samples was performed by LECO CS844 carbon and sulfur analyzer, LECO ONH836 oxygen analyzer, Hitachi S-3400N scanning microscope equipped with Thermo Scientific Noran System 7 EDS detector and WDS MagnaRay, X'Pert 3 Powder X-ray diffractometer and chemical “wet” analysis.
The grain composition analysis was carried out on woven screens with square mesh sizes: 0.5 mm; 1.0 mm; 2.0 mm; 3.0 mm and 5.0 mm. The dry material was screened on a vibrating mechanical device for 5 minutes (vibrations 300/min), after which particular fraction were weighed. A control sieving was then performed for 1 minut.
- g) p. 6, "Alundum retort capped at the bottom had a hermetic closure at the top, which was equipped with tubes for supplying reducing gas and receiving post-reaction gases (Fig. 1).". alumina reaction retort and alundum retort are the same device or two different devices? Because from figure 1 I can see only one furnace. Please check and if this is a repetition, adjust the text property
Obvious error, already removed, sorry.
- h) p. 7, "Ribbing in the form of longitudinal slats - blades mounted inside the retort". I think this is a repetition of the sentence above. Please check and correct accordingly
Again sorry, redundant text has been removed.
- i) p. 11, paragraph 3.3. "These research tests were performed with the use of two types of gas reducers: pure carbon monoxide and a gas mixture of- 50% CO + 50% H2. When using carbon monoxide, the reduction process was determined continuously, using automated analysis of CO and CO2 in the ABB Uras 14 gas analyzer. Thus, when using the mixture of CO and H2, the
course of reduction was observed and the amount of water condensing from the post-reaction gases was periodically recorded. After the reduction process, all samples weresubject to chemical analysis for the content of total iron (Fec) and metallic iron (Femet.). Furthermore, on selected samples, a structural analysis was performed for the content of metallic iron and other iron phases and compounds." This part should be moved in the methodology paragraph, because it describes a procedure and not a result
Reviewer’s suggestion has been implemented.
- j) p.13, "Investigations on reduction of iron oxides from iron-bearing waste materials with the". It would be better to divide the results section in two sub-paragraph. From here the paragraph related to CO+H2 reduction starts.
Currently, the research chapter is divided as it is seen below, and it seems that this division is both essentially and formally justified
- Research work and discussion
3.1. Research testing on reduction of iron oxides from iron-bearing waste materials in the Tamman furnace
3.2. Research tests on reduction of iron oxides from ferrous waste materials in a rotary furnace
- Studies on the reduction of iron oxides in waste materials by carbon monoxide
Mill scale reduction
Dust reduction
Sludge reduction
- Studies on the reduction of iron oxides in waste materials by the CO+H2 gas mixture
Mill scale reduction
Dust reduction
Sludge reduction
- Conclusions
- k) p.14, "During tests the course of reduction was observed and recorded by the amount of condensed water from post-reaction gases. Then, the reduced samples were subject to chemical analysis for the content of total and metallic irons and to structural analysis." already declared. Please, remove it
Text removed (was previously declared in chapter 2. Methodology of investigations
- l) p. 14, "by using as a reducing agent a mixture of reducing gases consisting
of 50% vol. of CO and "+ 50% vol. of H2". already declared. Remove it
Text removed.
- m) p. 14, table 16. why only two conditions were compared, if the Authors did several tests both in pure CO and CO+H2?
Comparative tests were carried out for selected conditions that gave a chance to obtain high degrees of reduction and metallization (eg. reduction temperature of 1000°C). Therefore, comparative reduction tests were also performed with increased flow of the reducer (Table 17).
- n) p. 15, "The difference in degrees of metallization of the reduced scale, depending on the reducer being applied was 9.4% in favor for the pure carbon monoxide used". Why? An explanation is mandatowy
The data included in the new table 16 show clearly the existing difference in the degree of metallization equal to 9.4%.
- o) p. 15, table 18. why there is not a comparison with pure CO reduction tests?
Unfortunately, it is not possible to compare reduction data with CO and mixture (CO+H2) due to inconclusive results of reduction experiments with gas mixture.
- p) p. 16, paragraph 3.3.1. What is done in the tamman refers to the proceeding presented in AISTECH conference. This part must be integrated in the paragraph 3.1.
Section 3.1 presents the most important results of the stationary layer reduction studies in the Tamman furnace. Unfortunately, I don't know what else should be integrated with this point, because this point is located in paragraph 3. "Research work and discussion" and only the related content should be placed there.
- q) p. 16, paragraph 3.3.2.
why the discussion did not take into account the test performed in CO+H2?
In addition, there is no explanation the different metallization ratios and reduction rates obtained in the different Samples
Indeed, there was no reference to the influence of the reducer quality on the reduction results and it was supplemented (p. 18), while regarding the remark "explanation the different metallization ratios and reduction rates obtained in the different Samples", I would like to explain that in this paper it was decided that the comments of the results should be of a more general nature, e.g. with regard to the influence of sample grain size, carbon content, temperature (based on the selected waste material), rather than the evaluation of individual samples. The research was assumed to be applied research.
- r) p.17, Table 21. These values should be put in the results section and not in the conclusions. Reduced samples in which atmosphere? CO or CO+H2?
New conclusions have now been developed.
Grammar errors
abstract: "These results indicate that in the case of sludge and scale, the degree of metallization meets the requirements for charge materials used in both blast furnace (BOF) and electric arc furnace (EAF) steelmaking processes, while in the case of reduced dust, this material can be used as enriched charge in the blast furnace process. Both pure carbon monoxide and a mixture of CO and H2 (50 vol.% CO + 50 vol.% H2) were used as a
reducer in the conducted research." please, check the font size because is different from the rest of the abstrakt
Error has been deleted.
- b) p. 2, "Metallurgical waste materials can be an alternative to conventional raw materials of construction". Replace of with for
Error has been deleted.
- c) p. 2, "by far exceeding other efec ts such as increased pressing force". effects
Error has been deleted.
- d) p. 10, "out are as follows: The". after : start with t
Error has been deleted.
- e) p. 11, "grain size, the basicity CaO/SiO2,". 2 as subscript
Error has been deleted.
- f) p.11, "Research tests on reduction of iron oxides in iron-bearing waste materials with the usage of carbon monoxide were carried out for all three groups of materials: dust, mill, scale and sludge." This sentence is redundant. Please delete
Error has been deleted.
- g) p. 17, "larger amount of redundant". Please correct redundant as reductant.
Error has been deleted.
Thank you for your insightful review and work to improve the quality of our publication.